# Electrophysiological Properties of Tetraploid Cardiomyocytes Derived from Murine Pluripotent Stem Cells Generated by Fusion of Adult Somatic Cells with Embryonic Stem Cells

**DOI:** 10.3390/ijms24076546

**Published:** 2023-03-31

**Authors:** Guoxing Xu, Azra Fatima, Martin Breitbach, Alexey Kuzmenkin, Christopher J. Fügemann, Dina Ivanyuk, Kee Pyo Kim, Tobias Cantz, Kurt Pfannkuche, Hans R. Schöler, Bernd K. Fleischmann, Jürgen Hescheler, Tomo Šarić

**Affiliations:** 1Center for Physiology and Pathophysiology, Institute for Neurophysiology, Faculty of Medicine and University Hospital Cologne, University of Cologne, 50931 Cologne, Germany; lf_xgx2006@hotmail.com (G.X.); afatima1@web.de (A.F.); alexei21@hotmail.com (A.K.); dina.ivanyuk@gmail.com (D.I.); kurt.pfannkuche@uni-koeln.de (K.P.); j.hescheler@uni-koeln.de (J.H.); 2Department of Physiology and Chinese-German Stem Cell Center, Tongji Medical College, Huazhong University of Science and Technology, Wuhan 430022, China; 3Institute of Physiology I, Life & Brain Center, Medical Faculty, University of Bonn, 53115 Bonn, Germany; mbreitba@uni-bonn.de (M.B.); bernd.fleischmann@uni-bonn.de (B.K.F.); 4Max Planck Institute for Molecular Biomedicine, Department of Cell and Developmental Biology, 48149 Münster, Germany; kee-pyo.kim@mpi-muenster.mpg.de (K.P.K.); cantz.tobias@mh-hannover.de (T.C.); h.schoeler@mpi-muenster.mpg.de (H.R.S.)

**Keywords:** embryonic stem cells, cardiac myocytes, diploid, polyploidy, tetraploid, cell fusion, reprogramming, electrophysiology, patch-clamp, heart regeneration

## Abstract

Most cardiomyocytes (CMs) in the adult mammalian heart are either binucleated or contain a single polyploid nucleus. Recent studies have shown that polyploidy in CMs plays an important role as an adaptive response to physiological demands and environmental stress and correlates with poor cardiac regenerative ability after injury. However, knowledge about the functional properties of polyploid CMs is limited. In this study, we generated tetraploid pluripotent stem cells (PSCs) by fusion of murine embryonic stem cells (ESCs) and somatic cells isolated from bone marrow or spleen and performed a comparative analysis of the electrophysiological properties of tetraploid fusion-derived PSCs and diploid ESC-derived CMs. Fusion-derived PSCs exhibited characteristics of genuine ESCs and contained a near-tetraploid genome. Ploidy features and marker expression were also retained during the differentiation of fusion-derived cells. Fusion-derived PSCs gave rise to CMs, which were similar to their diploid ESC counterparts in terms of their expression of typical cardiospecific markers, sarcomeric organization, action potential parameters, response to pharmacologic stimulation with various drugs, and expression of functional ion channels. These results suggest that the state of ploidy does not significantly affect the structural and electrophysiological properties of murine PSC-derived CMs. These results extend our knowledge of the functional properties of polyploid CMs and contribute to a better understanding of their biological role in the adult heart.

## 1. Introduction

Although most cells in an adult organism carry a diploid set of chromosomes, some tissues also contain polyploid cells [1]. In bone marrow (BM), platelet-producing megakaryocytes are a well-known example of obligate polyploid cells with the DNA content reaching multiplicities of the normal diploid state (up to 128n) [2]. Similarly, polyploidization and binucleation are characteristic features of liver growth and physiology. In the liver of the adult rat, about 70% of hepatocytes are tetraploid and 20% octaploid, and they can contain one or two nuclei [3]; in humans, approximately 30% of human hepatocytes are polyploid [4]. These percentages can vary during homeostasis, regeneration, and in various disease states [5,6,7,8]. In the heart, most ventricular cardiomyocytes (CMs) stop dividing shortly after birth and a large fraction of cells becomes polyploid [9]. In mice, about 80% of CMs have two diploid nuclei, while an additional 10% contain a single tetraploid nucleus [10,11,12,13]. In the adult human heart, the majority (54%) of CMs contain a single nucleus with a tetraploid DNA content [14,15]. However, their proportion can increase after myocardial infarction (MI) or pressure-overload disease [16,17,18,19], and in heart failure, some CMs can even attain a ploidy of 8n or 16n [20].

The physiological role of polyploidy has been extensively studied in recent years. In the heart, it is believed that the polyploid state is indicative of terminal differentiation and maturation of CMs [21], which may serve as a protection against excessive cell proliferation and tumor formation or to increase the functional capacity of the cell as a response to environmental stress, tissue injury, and disease [22,23,24]. Several studies in mice and zebrafish have suggested that induction of proliferation of pre-existing mononuclear diploid CMs predominantly contributes to heart regeneration after injury [12,25,26,27,28,29], leading to the suggestion that non-proliferative polyploid CMs cannot regenerate in the adult heart [12,24,26,30]. Hence, the elucidation of the molecular mechanisms of polyploidization and the characterization of molecular and functional properties of polyploid cells have been the main focus of a number of studies in recent years in the hope that this knowledge could be used to improve cardiac regeneration (reviewed in [24]).

Although the vast majority of endogenous polyploid CMs are formed by endoreplication, some polyploid CMs can also arise by the fusion of CMs with other types of somatic cells, especially in the context of exogenous cell transplantation for the treatment of MI [31,32,33,34,35]. Using genetic and lineage fate mapping approaches, several groups have demonstrated that transplanted BM-derived hematopoietic cells or mesenchymal stem cells (MSCs) can fuse with skeletal muscle cells [36], gastrointestinal cells [37], hepatocytes [38,39], neurons [40,41], and CMs [31,33,35,42,43]. However, the occurrence of cell fusion under these circumstances appears to be rare, and its contribution to tissue regeneration in physiological and pathological settings is low. Interestingly, the Sussman group has recently shown that cardiac progenitor cells (CPCs) and MSCs, fused in vitro prior to transplantation, exhibited enhanced reparative potential in a mouse model of MI relative to individual stem cells or combined cell transplantation [44]. These results suggest that combining the beneficial properties of different cell types in vitro by fusion into a single tetraploid cell entity may represent a strategy for enhanced cardiac regeneration.

Cell fusion also represents a useful tool for investigating the mechanism of cellular plasticity and reprogramming [45,46,47]. In vitro fusions between embryonic stem cells (ESCs) or induced pluripotent stem cells (iPSCs) and various somatic cell types, such as splenocytes, thymocytes, fibroblasts, neural cells, or myeloid precursor cells, have been reported to promote the epigenetic reprogramming of the somatic cell genome to a pluripotent embryonic phenotype [48,49,50,51,52,53,54]. The resulting fusion hybrids, despite being tetraploid or near-tetraploid, were morphologically indistinguishable from normal ESCs. They exhibited a transcriptional and epigenetic profile similar to that of ESCs, with the potential to differentiate into multiple lineages in vitro, form teratomas in immunodeficient animals, and contribute to all three primary germ layers of chimeric embryos. Fusion hybrids of fibroblasts and murine ESCs were even shown to contribute to several tissues of liveborn adult chimeric mice, containing cells with near-tetraploid karyotype reminiscent of hybrid cells injected into the diploid blastocysts [55]. Studies examining the ploidy of ESC/somatic cell hybrids using fluorescent and microsatellite marker analysis and in situ hybridization have revealed that hybrid clones exhibit variable chromosome numbers ranging from 40 to 85 due to extensive loss or gain of individual chromosomes [48,55,56]. However, near-tetraploid genomes in fusion hybrids were stably retained, and complete segregation of parental somatic and ESC genomes did not occur even after prolonged cell passaging [56] or differentiation [57]. Collectively, these data suggest that fusion-derived cells can be propagated as stable pluripotent stem cell (PSC) lines capable of supporting in vitro differentiation, initial stages of embryonic development, and even tissue growth in an adult organism.

Although the biological role of polyploid CMs under physiological and pathological conditions is gradually being elucidated, the electrophysiological properties of diploid and polyploid CMs have only scarcely been characterized. In this study, we generated stable tetraploid PSC lines by in vitro fusion of murine diploid somatic cells and PSCs to obtain homogeneous populations of tetraploid CMs at sufficient quantities for studying their electrophysiology. Comparative analyses of functional properties of diploid ESC-derived and near-tetraploid fusion-hybrid (FH) PSC-derived CMs showed that the two types of CMs have similar characteristics, suggesting that the state of ploidy alone does not significantly affect the electrophysiological properties of murine PSC-derived CMs. Although immature tetraploid CMs arising from fusion-derived PSCs may differ in their properties from adult native polyploid CMs, the findings reported here offer valuable insights into the functional effects of polyploidy in CMs and may contribute to a better understanding of the physiological role of polyploidy in the adult heart.

## 2. Results

### 2.1. Generation of Hybrid Cells by Fusion of ESCs with Adult Somatic Cells

In order to facilitate the selection of somatic cells reprogrammed to a pluripotent state by fusion with ESCs, we chose the hypoxanthine–guanine phosphoribosyl transferase (HGPRT)-deficient ESC line HM-1, derived from the 129/Ola mouse strain carrying the major histocompatibility complex (MHC) class I haplotype H-2^b^, to serve as a PSC fusion partner with somatic cells from BM or spleen of wild type DBA/2J mice carrying the MHC class I H-2^d^ haplotype (Figure 1A). While the hypoxanthine-aminopterin-thymidine (HAT) selection medium is toxic to HM-1 cells, hybrid PSCs generated as a result of the fusion between an ESC and a somatic cell can survive in this medium. Using this approach, we have established several stable HAT-resistant clones with BM cells and splenocytes. In control fusion reactions containing only ESCs or somatic cells, no ESC-like colonies could be retrieved after selection in the HAT-containing medium. All FH clones that were established in fusions between ESCs and somatic cells exhibited the colony morphology indistinguishable from that of the original HM-1 ESC line, and these clones could be propagated for at least 20 passages without losing their typical ESC-like appearance (Figure 1B). One BM-derived (FH-2.1) and one splenocyte-derived clone (FH-4.2) were randomly selected for further studies.

### 2.2. DNA Content and Cytogenetic Analysis

The determination of the DNA content by flow cytometric analysis of propidium iodide (PI)-stained cells revealed that the majority of cells in both FH clones carry tetraploid (4n, cells in G0/G1 phase of the cell cycle) and octaploid (8n, cells in the G2/M phase of the cell cycle) nuclei (Figure 1C). In contrast, parental HM-1 ESCs (Figure 1C) and somatic cells showed only the DNA staining pattern expected for diploid cells (2n and 4n nuclei). To further confirm the polyploid character of fusion-derived PSCs, we compared the chromosomal content of FH cells and the parental HM-1 ESCs by routine cytogenetic analysis of Hoechst-stained metaphases (Appendix A). While the modal number of chromosomes within spreads of HM-1 cells was slightly aneuploid (mode = 42), the undifferentiated FH cells were more heterogeneous with chromosome numbers in the majority of cells ranging, for FH-2.1 and FH-4.2 clones, from 68 to 76 (mode = 73) and 67 to 71 (mode = 70), respectively (Figure 1D). Cytogenetic analyses performed with cells isolated from embryoid bodies (EBs) on day 10 of differentiation showed that differentiated HM-1 and FH cells had almost the same number of chromosomes as their undifferentiated counterparts (Figure 1D), indicating that the near-tetraploid state in FH cells was stably maintained in the course of differentiation.

### 2.3. Multicolour Fluorescence In Situ Hybridization (mFISH)

In order to more accurately assess the chromosomal composition of HM-1 and FH cells, both FH cell lines and HM-1 ESCs were analyzed by mFISH with painting probes specific for all 21 different mouse chromosomes. As expected, cell lines FH-2.1 and FH-4.2 turned out to be pseudotetraploid, and HM-1 cells were diploid. Detailed analyses of metaphases for rearrangements and changes in chromosome numbers revealed that ten metaphases analyzed for HM-1 ESCs revealed that this cell line is diploid without any structural aberrations. Three cells had a normal male karyotype (2n = 40, XY), one cell lost a chromosome 1, which might be a spreading artifact, four cells gained a chromosome 8 (2n = 41, XY, +8), and two cells had gained a chromosome 8 and a chromosome 11 (2n = 42, XY, +8, +11) (Appendix A). Trisomies of chromosomes 8 and 11 are known to be the most common aberrations in mouse ESC lines [58].

The FH-2.1 and FH-4.2 cells were pseudo-tetraploid, carrying 63, 69, 73, 73, 87, and 67, 70, 72, 72, 89 chromosomes, respectively, in the five metaphases that were karyotyped. In both cell lines, one of five metaphases had no Y chromosome, three cells had only one Y chromosome each, and one cell had two Y chromosomes. No structural aberrations were detected in the five FH-2.1 and four FH-4.2 metaphases analyzed (Appendix A). However, one FH-4.2 cell had two translocation chromosomes, one consisting of segments of chromosomes 10 and 15 (der10(10;15)), and the other translocation consisting of segments of chromosomes 4 and 14 (der4(4;15)). Additionally, there was an enlarged chromosome 4 (dup(4)) (Appendix A). These data further confirm that fusion-derived PSC lines contain a tetraploid or near-tetraploid set of chromosomes.

### 2.4. SNP Genotyping of Fusion-Derived Cells

To enable the tracing of chromosomes from both types of fusing cells, we chose fusion partners from different mouse strains that can be distinguished by their single nucleotide polymorphisms (SNPs): HM-1 ESCs were from 129/Ola and somatic cells from DBA/2J mouse. The reference variant alleles of the SNPs arbitrarily selected for each chromosome are tabulated in Appendix A. In order to determine if both parental chromosome complements are present in FH cells, the genomic DNA isolated from somatic, ESC, and FH cells was amplified by PCR and sequenced using the primers for a DNA segment carrying one SNP variation on each of the 19 murine autosomes. Figure 2A illustrates the result of the sequencing of the SNP on chromosome 7, demonstrating the presence of only one polymorphic allele in HM-1 ESCs (nucleotide A) and somatic cells (nucleotide C), as expected at this position, for each genotype. In contrast, at this position, FH cells contained the variant alleles of both of their parental cell lines (nucleotides A and C). The same results were obtained for all other chromosomes tested (Appendix A) and confirmed in at least two independent experiments. The presence of both alleles in both FH cell populations demonstrated that the fusion clones retained both parental genomes and at least one of each pair of parental autosomes and that they were not lost in the course of cell expansion.

### 2.5. MHC Class I Haplotyping of Fusion-Derived Cells

The parental cells used for fusion were chosen to differ in their MHC class I haplotypes (HM-1 ESCs have H-2^b^ and somatic cells H-2^d^ haplotype), enabling further confirmation and monitoring of the hybrid character of fusion-derived PSCs by flow cytometry and immunocytochemistry. We and others have previously shown that murine ESCs do not express detectable levels of MHC class I molecules on their surface in an undifferentiated state and that their expression cannot be induced by interferon-γ (IFNγ) [59,60] (see also Appendix A). In contrast, ESC derivatives express MHC class I molecules at low but detectable levels, and IFNγ can increase this expression. Accordingly, undifferentiated FH cells also did not express notable amounts of MHC class I molecules and were not responsive to IFNγ (Figure 2B, upper panels). In order to determine if both parental MHC haplotypes can be detected on the surface of FH cells, they were differentiated into embryoid bodies (EBs) and analyzed on day 4 of differentiation with and without IFNγ-treatment. Untreated EB cells of FH-2.1 and FH-4.2 clones expressed low but detectable levels of both MHC class I haplotypes, and IFNγ increased their expression (Figure 2B, lower panels). These data provide additional evidence for a hybrid character of FH cells and show that all MHC class I alleles originating from the parental cells are active in FH cells and respond to exogenous stimuli.

### 2.6. Pluripotency of FH Cells

Next, we asked whether FH cells possess other properties of PSCs besides exhibiting ESC-like colony morphology. Both FH clones expressed the pluripotency marker Oct4 and SSEA-1, as demonstrated by immunocytochemistry and flow cytometry, respectively (Figure 3A,B), and exhibited robust demethylation of CpG dinucleotides within the promoter regions of pluripotency-associated genes *Oct4* and *Nanog* (Appendix A). In addition, these cells formed teratomas in immunodeficient mice; these tumors were comprised of derivatives of all three germ layers (Figure 3C and Appendix A). Furthermore, to elucidate if splenocyte-derived FH-4.2 cells lost markers of the somatic cell partner as a result of unidirectional reprogramming towards pluripotency, we assessed the expression of markers of the lymphoid and myeloid lineages that are expressed on splenocytes. Using splenocytes as positive control cells, we found that these cells express antigens specific for leukocytes (CD45, 97%), monocytes/macrophages (CD11b, 35%), B lymphocytes (CD19, 57%), and T lymphocytes (CD3a, 39%), respectively (Figure 3D, upper panels). However, undifferentiated HM-1 ESCs did not express any of these markers (Figure 3D, middle panels). Undifferentiated FH-4.2 cells also did not express significant levels of CD45, CD11b, and CD19 antigens, but low levels of T cell antigen CD3a were expressed in 2.4% of the cells (Figure 3D, lower panels). Comparable results were obtained for markers of bone marrow cells, which were neither expressed on HM-1 ESCs nor on the FH-2.1 PSCs (Appendix A). These results demonstrate that the somatic genome in FH-PSCs is dominated by chromatin-modifying factors expressed in ESCs, leading to the silencing of somatic cell genes in FH cells and thereby establishing the pluripotent state.

### 2.7. Cardiac Differentiation of FH Cells

In the course of differentiation, cells of the clones FH-2.1 and FH-4.2 formed typical EBs, which were somewhat smaller than ESC-derived EBs. After plating on gelatine-coated dishes, EBs derived from fusion clones also revealed a lower surface adhesion capacity compared to their ESC-derived counterparts. All cell types displayed a high cardiac differentiation efficiency of 90–100%, assessed by the percentage of EBs with spontaneously contracting areas (Figure 4A). However, EBs from clones FH-2.1 and FH-4.2 reached the maximal percentage of beating EBs later (days 11–14) than ESC-derived EBs (days 8–9). During differentiation from day 8 to day 25, the beating frequency of spontaneous contraction was in the range of 55–75 min^−1^ in all cell types, and it did not significantly differ between ESC and FH cell-derived EBs nor change over time (Appendix A).

To further prove the cardiac differentiation potential of FH cells, we analyzed the expression of the cardiac marker genes alpha-myosin heavy chain (*αMHC*), ventricular form of myosin light chain (*MLC2v*), and cardiac troponin T (*cTnT*) in undifferentiated cells and EBs at day 10 and day 16 of differentiation via RT-PCR. *αMHC* could not be detected in undifferentiated FH cells and day 10 EBs, but was strongly induced in EBs of both FH clones at day 16 of differentiation (Figure 4B). *MLC2v* and *cTnT* showed weak expression in undifferentiated FH cells but were strongly upregulated in FH-derived EBs at day 16 of differentiation. Interestingly, the *MLC2v* mRNA levels in day 10 EBs were significantly higher in the FH-2.1 than in the FH-4.2 clone, suggesting that some interline differences in CM differentiation may exist. Immunocytochemical staining of ESC-derived and FH-PSC-derived EBs revealed that the beating cells in all these structures expressed the cardiac proteins sarcomeric α-actinin and cTnT and exhibited the cross-striation pattern typical for CMs (Figure 4C). These analyses also revealed that FH-PSC-derived CMs express the MHC class I molecules of both parental fusion partners (H-2K^b^ from ESCs and H-2K^d^ from somatic cells), further indicating the retention of both parental genotypes in these differentiated FH cell-derivatives (Figure 4D; see also Appendix A for validation of antibodies used for staining of the H-2K^b^ and H-2K^d^ molecules).

### 2.8. Characteristics of Spontaneous Action Potentials (APs) in Fusion-Derived CMs

Using the whole-cell patch-clamp recording technique, we first determined the membrane capacitance. These analyses showed that the membrane capacitance of CMs derived from diploid HM1 ESCs and near-tetraploid FH-2.1 and FH-4.2 PSCs does not differ significantly (Appendix A). To characterize different phenotypes of FH-PSC and ESC-derived CMs, we recorded and analyzed spontaneous APs on days 16–19 of differentiation. In CMs, irrespective of their fusion or ESC origin, the AP types representative of mostly atrial- and ventricular-like cells were found at this stage of differentiation, while some APs resembled pacemaker-like cells, while others could not be clearly classified (Figure 5). Among them, most cells were of ventricular (52% in ESC-derived CMs) or atrial-like type (46% of FH-2.1 CMs and 54% of FH-4.2 CMs) (Table 1). Except for the statistically significant difference in “Peak” and “Vdd” parameters between atrial-like FH-2.1 and FH-4.2-CMs, no other statistically significant differences were found in all other AP parameters (MDP, AP-height, AP-frequency, Vmax, APD90, and APD90/APD50) between diploid and near-tetraploid CMs within each CM-subtype (Table 1 and Appendix A). However, statistically significant differences were found between different CM-subtypes for MDP (P-like vs. A-like), AP height (P-like vs. A-like and P-like vs. V-like), Vmax (P-like vs. A-like and P-like vs. V-like), APD90 (P-like vs. V-like and A-like vs. V-like), and APD90/APD50 ratio (A-like vs. V-like) (see Appendix A).

### 2.9. Effects of Muscarinic and β-Adrenergic Receptor Agonists on AP Frequency

Carbachol (CCh), a synthetic acetylcholine analogon, was applied to investigate the muscarinic signaling in CMs. This parasympathomimetic drug reduced the AP frequency in HM-1 CMs by 59 ± 9% (n = 11), but this effect was not statistically significant (*p* > 0.05; Figure 6A,B). However, the reduction of AP frequency in FH-2.1 CMs (82 ± 5%, n = 9, *p* < 0.01) and FH-4.2 CMs (47 ± 8%, n = 14, *p* < 0.05) was statistically significant, and these negative chronotropic effects were reversible upon washout (Figure 6A,B). We also examined the response of CMs to β-adrenergic regulation with isoproterenol (Iso). When 1 μM Iso was administered to CMs, the beating frequency increased significantly only in FH-2.1 CMs (111 ± 47%, n = 7, *p* < 0.01), while the increase in HM-1 CMs (53 ± 18%, n = 12) and FH-4.2 CMs (41 ± 13%, n = 13) was not statistically significant (*p* > 0.05; Figure 6C,D). There were no significant differences between the effects on CMs derived from diploid ESC and near-tetraploid FH-PSC lines (Figure 6B,D). Positive chronotropic effects of Iso were partially reversible upon washout, but the values after washout did not reach statistical significance compared to drug treatment.

### 2.10. Effects of Ion Channel Blockers on Spontaneous APs

Spontaneous APs resulted from the opening and closing of voltage-gated ion channels in the sarcolemmal membrane. Therefore, we applied lidocaine as a Na^+^ channel blocker, nifedipine as an L-type Ca^2+^ channel blocker, and E4031 as a K_r_ (mERG) channel blocker to analyze their functional expression in fusion-PSC and ESC-derived CMs. The spontaneous beating rates were found to be in the range of 170 to 240 bpm and varied strongly in different groups from cell to cell (Figure 7). All blockers led to negative chronotropic effects in all groups of CMs. Lidocaine decreased the AP frequency or abolished the spontaneous beating in most CMs. As a result, mean AP frequency significantly decreased by 91 ± 20% in HM-1 CMs (n = 6, *p* < 0.05), 84 ± 10% in FH-2.1 CMs (n = 7, *p* < 0.05 after washout), and 71 ± 10% in FH-4.2 CMs (n = 8, *p* < 0.05) (Figure 7A,B). There was no statistically significant difference between beating frequency in CMs derived from FH-PSC and ESC lines. Nifedipine led to a complete abrogation of spontaneous APs in all ESC-CMs (n = 8), FH-2.1 CMs (n = 6), and FH-4.2 CMs (n = 7) (Figure 7C,D). Treatment of HM-1, FH-2.1, and FH-4.2 CMs with E4031 caused a reduction of the mean AP frequency by 44%, 54%, and 67%, respectively, and many CMs stopped beating under the application of this drug. This AP frequency decrease was significant in FH-4.2 CMs (n = 8, *p* < 0.05) but not in the other two CM groups (Figure 7E,F). The effects of lidocaine, nifedipine, and E4031 were all partially reversible upon washout (Figure 7B,D,F). The comparison of drug effects on CMs derived from diploid ESC and near-tetraploid FH-PSC lines revealed no significant differences between these cell types.

### 2.11. Expression of Cardiac-Specific Voltage-Gated Na^+^ and L-Type Ca^2+^ Channels

For further electrophysiological characterization, we performed whole-cell voltage-clamp on FH-PSC- and ESC-derived CMs. Typical currents through voltage-gated Na^+^ and L-type Ca^2+^ channels are presented in Figure 8. To estimate the expression of functional ion channels in the cell membrane, we calculated current densities by normalizing the maximal current amplitude to the cell size. These analyses showed that Na^+^ current density in FH-2.1 CMs (n = 24) and FH-4.2 CMs (n = 25) was similar to that in ESC-CMs (n = 45) (Figure 8A,B). Ca^2+^ channel density was significantly (*p* < 0.05) higher in FH-4.2 CMs (n = 10) as compared to HM-1-ESC-derived CMs (n = 18), but CMs derived from the FH-2.1 clone had a Ca^2+^ channel density similar to that in ESC-CMs (Figure 8C,D). Voltage-dependence of activation of Na^+^ and L-type Ca^2+^ currents were mostly similar in all FH-PSC- and ESC-derived CMs (Figure 8A,C; insets).

## 3. Discussion

In this study, we describe the successful reprogramming of murine somatic cells from bone marrow and spleen by fusion with murine ESCs and perform an extensive comparison of the functional properties of near-tetraploid and diploid CMs derived from pluripotent FH cells and parental ESCs, respectively. Somatic and ESC fusion partners were chosen to originate from different strains of mice that could be distinguished based on their specific MHC class I haplotypes and single nucleotide polymorphisms. These features facilitated the subsequent confirmation of the fusion character of reprogrammed cells and their differentiated derivatives. The generated fusion-derived PSCs could be propagated in culture for over 20 passages and exhibited an ESC-like morphology, expressed pluripotency markers, epigenetically reprogrammed promoter regions of pluripotency-associated genes *Oct4* and *Nanog*, downregulated somatic cell lineage markers, and MHC class I molecules and formed teratomas in immunodeficient animals. Thus, FH cells generated in this study possess similar pluripotent properties as those reported for hybrid PSCs in previous studies [48,49,53,54,55,56].

Using ploidy and karyotype analyses, as well as the SNP genotyping and MHC class I immunophenotyping, we show that the FH cells investigated in this study contain both parental genomes. However, the number of chromosomes in most cells was lower than expected for a complete tetraploid genome, indicating that a loss of chromosomes took place in FH-PSC clones. According to previous reports, chromosome loss most likely occurs in the early phase of establishing viable hybrids until the reprogramming of the somatic genome is completed [56]. This is in agreement with our observation that the near-tetraploid state of the FH cells was stably maintained in hybrid cells during prolonged in vitro cultivation. Moreover, FH cells at day 10 of differentiation had a similar modal chromosome number to undifferentiated FH cells, suggesting that the karyotype of FH cells was preserved in the process of differentiation. The detection of MHC class I molecules of both parental haplotypes on the surface of FH cell-derived CMs further confirms the cytogenetic stability of hybrid cells during differentiation. Other groups have also found that hybrid cells maintain their tetraploid or near-tetraploid chromosome complements during in vitro expansion [54,56] and differentiation [57]. The Serov group reported that near-tetraploid hybrid ESCs were even capable of contributing to the development of multiple organs and tissues in chimeric embryos and adult mice [55]. Moreover, using cytogenetic and microsatellite analyses, they showed that cells derived from chimeric embryos or adult chimeras still contained the initial near-tetraploid karyotype of original hybrid cells that were injected into diploid blastocysts. These remarkable findings indicate that hybrid cells efficiently retain their chromosomal stability even during in vivo development and that the developmental potential of hybrid cells is not significantly affected by their ploidy, which was also observed by others [52,61,62]. In general, increased ploidy is associated with an increase in cell size as well as mRNA and protein abundance [63]. However, in our experimental model, we do not observe clear effects of ploidy on membrane capacitance. One possibility is that a clear effect of ploidy on cell size may have been masked by the presence of different CM subtypes as well as CMs at different stages of maturation in these cultures, potentially contributing to the observed variability in membrane capacitance.

In agreement with the note that hybrid cells possess unperturbed developmental potency, FH cells generated in this study differentiated to CMs to a similar extent as parental ESCs, albeit with somewhat slower kinetics. Other groups have also reported that hybrid cells can give rise to spontaneously beating CMs [48,49], but detailed functional analyses of these cells have not yet been performed. Here, we demonstrate that CMs derived from FH cells are responsive to β-adrenergic and muscarinic regulation [63,64]. Additionally, using the specific blockers of I_CaL_ (nifedipine) [65,66,67], I_Na_ (lidocaine) [68,69], and I_Kr_ (E4031) [70,71,72], we show that these important cardiac ion channels are expressed and are functional in FH cell-derived CMs. The L-type Ca^2+^ channel blocker nifedipine caused the halt of the AP in FH-PSC-derived CMs to a similar extent as in ESC-derived CMs, and this is in agreement with previous studies [73,74]. Lidocaine led to a decrease in AP frequency and V_max_, implying the important role of voltage-gated Na^+^ channels in forming the upstroke of AP in the brief phase of depolarization [75,76]. E4031 caused MDP depolarization, suggesting the I_Kr_ contribution to the MDP balance [71,77,78,79]. All these findings were similar to those obtained with ESC-CMs and indicated that the ploidy state does not exert a significant effect on the functional properties of CMs and their maturation state. However, the finding of a significantly higher density of L-type Ca^2+^ channels in FH-4.2 CMs compared to FH-2.1- and ESC-derived CMs suggests that these cells could differ in some properties and that these are probably not caused by polyploidy but by other factors, such as by differences in gene expression due to a variable effect of reprogramming on different somatic cells of origin.

Despite many similarities between the FH-PSC and ESC-derived CMs, there were some other minor differences between the clones. First, FH-2.1 CMs and FH-4.2 CMs displayed a slightly altered differentiation process compared to ESC-CMs, as they showed a later onset of spontaneous beating, a smaller EB size, and a lower surface adhesion capability. Delayed maturation has also been reported for CMs derived from some iPSC lines [63,80], suggesting similarity in the time course of differentiation between these different reprogramming approaches. However, these differences may simply be due to variability in the differentiation potential of different PSC lines and not necessarily, an FH or iPSC-specific phenomenon [81,82,83,84]. Secondly, our results imply that FH-2.1 and FH-4.2 CMs predominantly develop into CMs of atrial-like phenotype, whereas CMs derived from the HM-1 ESC line mostly revealed ventricular-like APs. This different cell fate might result from different reprogramming approaches or inherent variability between different PSC lines [84,85], and might reflect a different manifestation of somatic memory in fusion-PSC- and ESC-derived CMs [86,87]. Further studies have to show whether these minor differences are due to the polyploid nature of FH-PSC-derived CMs and whether human polyploid CMs derived from tetraploid PSCs also have properties similar to their diploid PSC counterparts. Our findings cannot be directly extrapolated to native CMs because the electrophysiological properties of native adult CMs and CMs derived from ESCs or iPSCs differ significantly due to different levels of their structural and functional maturity [88]. Therefore, the electrophysiological properties of diploid and polyploidy CMs isolated from the hearts of mice, as well as humans, should be performed in future studies.

Additional analyses, such as a subsequent single-cell gene expression analysis, could have provided some additional confirmation for our electrophysiological classification of the measured cells. However, it is unclear whether additional biochemical or genetic data would help identify different CM subtypes unequivocally. For example, Yechikov and coworkers have correlated the action potential profiles of individual pacemaker-, atrial-, and ventricular-like hiPSC-CMs with the expression of the proposed pacemaker-specific markers HCN4 and Isl1 at the protein level in the same cell [89]. They found that these two markers were expressed in all three hiPSC-CM subtypes. Therefore, these markers alone are not sufficient to identify hiPSC-derived pacemaker-like CMs. In another study using single-cell RNA-sequencing, the group of Timothy Nelson has shown that individual cellular expression patterns alone are not able to categorize the individual hiPSC-CMs into atrial, ventricular, or nodal subtypes, as determined by electrophysiology measurements [90]. Most recently, Schmid and coworkers used electrophysiological and single-cell RT-qPCR data from the same cell to determine if ion channel transcripts correlate with the electrophysiological parameters [88]. They showed that the majority of individual cells in three different commercially available hiPSC-CM preparations did not represent chamber-specific cell populations present in the adult human heart and exhibited unexpected combinations of ion channel transcripts and electrophysiological parameters on the single-cell level, combining characteristics of nodal, atrial, and ventricular CMs. Therefore, electrophysiological classification of CMs can be considered an acceptable method to confirm the presence of different CM subtypes in cultures.

Although the biological role of polyploidy in CM has been intensively studied in recent years, knowledge about the impact of increased ploidy on the functional properties of CM is limited [22]. The first study addressing this question was performed on mononucleated and binucleated CMs isolated from adult rabbits’ left atrium and pulmonary veins; it showed that they have different electrical activities and calcium dynamics [91]. This study reported that mononucleated native CMs had a more positive resting membrane potential, faster beating rates, larger calcium transients, as well as a lower inward-rectifier potassium channel Kir2.3 and higher ryanodine receptor 2 (RyR2) densities than binucleated CMs, suggesting that CM ploidy may contribute to this different electrophysiology in the left atrium and pulmonary vein CMs. The second study compared the functional properties of mononuclear diploid human ESC-derived ventricular CMs and their multinuclear counterparts generated by a polyethylene glycol-induced fusion of a 1:1 mixture of human ESC-derived CMs genetically engineered to express GFP or tdTomato under the control of a cardiac ventricular-specific *MLC2v* promoter [92]. These authors found that the beating frequency of mononuclear CMs was lower, and the AP duration was slightly longer in fusion-derived ESC-CMs compared to their mononuclear controls. However, no significant differences were observed in calcium handling, mitochondrial biogenesis, and viability between these CM populations.

Similar contradictory results have also been reported for transcriptional profiles of native diploid and polyploid CMs. Yekelchyk and coworkers showed that the transcriptional profile of adult murine mono- and multi-nucleated CMs was highly similar, as determined by single-cell RNA-sequencing [93]. Hesse et al. also showed that gene expression signatures in healthy CMs, as well as in CMs in the marginal zone of the lesion after MI, were independent of the number of nuclei or their ploidy [20]. In contrast, Windmueller et al. found that murine mono- and binucleated CMs analyzed at various times of development were transcriptionally distinct [94]. This inconsistency in the results published so far makes it clear that further studies are needed to conclusively answer the question of the molecular and functional properties of diploid and polyploid CMs.

In this study, we demonstrate that near-tetraploid PSCs derived by the fusion of ESCs with somatic cells give rise to functional CMs that have established hormonal regulation pathways and functionally expressed cardiac ion channels with characteristics very similar to those of diploid ESC-derived CMs. Further studies are required to determine the functional properties of human diploid and polyploidy CMs. Since the accessibility of native human CMs is limited due to ethical reasons, the cell fusion approach described here to generate polyploid PSCs could be applied to the human system to establish an unlimited source of polyploid human CMs for molecular and functional studies. It is also worth noting that tetraploid cells are capable of contributing to the growth of apparently healthy tissues in an adult organism, including the heart [55], and that hybrids produced by the fusion of CPCs and MSCs had a higher regenerative potential in a mouse model of MI than individual stem cells or combined cell transplantation [44]. Therefore, it is tempting to propose that CMs derived from tetraploid PSCs with or without additional genetic enhancement could also be considered and tested as an alternative source of cells for cardiac repair if they offer the desired safety, functionality, and efficacy after transplantation.

## 4. Materials and Methods

### 4.1. ES Cells

A hypoxanthine–guanine phosphoribosyl transferase (HGPRT)-deficient ESC line HM-1 [95], derived from 129/Ola male mice with normal diploid karyotype and major histocompatibility complex (MHC) class I haplotype H-2^b^, was chosen as a PSC fusion partner. The HM-1 cells were cultured in cell culture dishes coated with 0.1% gelatine (Sigma-Aldrich, St. Louis, MO, USA) in Dulbecco’s modified Eagle’s medium (DMEM) supplemented with 15% fetal bovine serum (FBS), 100 U/mL penicillin, 100 µg/mL streptomycin, 1% non-essential amino acids (NEAA), and 0.1 mM β-mercaptoethanol (βME, all from Invitrogen GmbH, Karlsruhe, Germany), and 1000 U/mL LIF (ESGRO, Millipore, Billerica, MA, USA) at 37 °C in 5% CO_2_ and high humidity. The cells were passaged every two or three days.

### 4.2. Somatic Cells

In order to facilitate validation of successful cell fusions, the somatic cells originated from inbred female mice of the DBA/2J strain (Jackson Laboratory, Bar Harbor, ME, USA), which have normal diploid karyotype and express MHC class I haplotype H-2^d^. Animals were sacrificed by cervical dislocation, the BM was flushed out of tibiae and femora using phosphate-buffered saline (PBS, pH 7.4), and the spleen was removed from the abdominal cavity and homogenized in PBS. Both resulting cell suspensions were filtered through a 100 µm cell strainer, washed with PBS, and counted.

### 4.3. Cell Fusion

Splenocytes and mononuclear BM cells were fused with HM-1 cells according to the previously published protocol [96]. Briefly, 1.6 × 10^6^ mononuclear BM cells were mixed in a 50 mL Falcon with 4 × 10^5^ HM-1 ESCs and 5 × 10^6^ splenocytes with 4 × 10^5^ HM-1 ESCs. Both suspensions were treated as follows: cells were pelleted, pellets dissociated by adding, dropwise, 1 mL of prewarmed 50% polyethylene glycol 1500 over the time period of 1 min, then further diluted with 2, 3, 5, and 10 mL of DMEM without supplements (each for 1 min) with constant stirring at 37 °C and 300 rpm in a Thermomixer (Eppendorf, Hamburg, Germany) equipped with a 50 mL tube adapter. Cells were then pelleted by centrifugation at 130× *g* for 10 min at RT in a swing-out rotor, resuspended in 200 µL complete ESC medium containing 1000 U/mL LIF, and incubated at 37 °C for 20 min. Thereafter, cells were seeded onto murine embryonic fibroblasts (MEFs) inactivated by irradiation and cultured with medium changes after 24 h and then every other day. To eliminate unfused ESCs, HAT supplement (50×, Invitrogen) was added 48 h later into the medium to a final concentration of 2% (0.1 mM hypoxanthine, 0.4 μM aminopterin, and 0.016 mM thymidine). Clones were picked after 6 days and subcloned by culturing in the presence of 2% HAT in ESC medium. Five clones of BM-derived and three clones of splenocyte-derived clones could be established, and fusion-derived hybrid (FH) clones termed FH-2.1 (derived from BM) and FH-4.2 (derived from splenocytes) were chosen for further validation.

### 4.4. Cardiac Differentiation

Cardiac differentiation of FH clones and HM-1 ESCs was performed using the hanging drop method [74]. Differentiation medium consisted of Iscove’s modified Dulbecco’s medium (IMDM), supplemented with 20% FBS, 100 U/mL penicillin, 100 µg/mL streptomycin, 1% NEAA, and 0.1 mM βME, then, 20 µL drops containing 400 cells each were placed on the lids of cell culture dishes filled with PBS. After 2 days of cultivation in hanging drops, the embryoid bodies (EBs) were transferred into a differentiation medium and cultivated for a further 5 days in suspension. On day 7, EBs were plated onto 0.1% gelatine-coated cell culture dishes for further differentiation. On day 8 or 9, beating areas were observed in some EBs. Beating frequencies were determined starting from day 8, and CMs were isolated on days 10–19 of differentiation for further analyses. For patch-clamp experiments, beating areas were cut out, dissociated by 0.25% Trypsin/EDTA treatment, and then plated on fibronectin-coated (2.5 μg/mL) 20 × 20 mm glass cover slips in 3.5 cm dishes. Cells were incubated for 36–72 h prior to measurements.

### 4.5. Karyotype Analysis

Cells were arrested in metaphase for 2 h by the addition of 0.1 µg/mL Demecolcin (Sigma-Aldrich) into the culture medium. Then, cells were trypsinized, washed in PBS, and pelleted by centrifugation. Pellets were gently resuspended by shaking and the dropwise addition of 1 mL of 75 mM potassium chloride solution (KCl). Cells were further diluted with 4 mL 75 mM KCl and incubated for 15 min at room temperature. Thereafter, 1 mL of chilled (−20 °C) fixative (methanol:acetic acid 3:1) was added and cells were pelleted by centrifugation at 130× *g* for 5 min. The pellet was gently resuspended in 1 mL fixative and incubated for 30 min at room temperature. Cells were washed twice and resuspended in 1 mL fixative. Finally, the cell suspension was dropped onto chilled glass slides and air-dried. DNA was stained using Hoechst 33342 (2 µg/mL, Sigma-Aldrich). Images of metaphase chromosomes were acquired on an Axiovert 200M fluorescence microscope equipped with an AxioCam MRm and analyzed with Axiovision 4.5 software (Carl Zeiss Microimaging, Oberkochen, Germany). Multicolor fluorescence in situ hybridization (mFISH) analysis was carried out as a service by CHROMBIOS (Nussdorf, Germany).

### 4.6. Flow Cytometry

To determine the expression levels of somatic cell markers on FH cells, cells were dissociated by trypsinization and incubated for 30 min at 4 °C with phycoerythrin (PE)- or fluorescein isothiocyanate (FITC)-conjugated antibodies against murine CD45, CD11b, CD19 (Caltag Laboratories, Hamburg, Germany), and CD3a (BD Pharmingen, Heidelberg, Germany). The expression of cell surface MHC class I molecules was assessed by using FITC-conjugated anti-mouse H-2K^b^ and PE-conjugated anti-mouse H-2K^d^ antibodies (BD Pharmingen). FITC or PE-conjugated mouse IgG2a antibodies served as isotype controls. Detection of PE and FITC labeled cells was accomplished on a FACScan flow cytometer equipped with a 488-nm argon laser (Becton Dickinson, Heidelberg, Germany) using BD CellQuest software (v3.5). WinMDI 2.8 software was used to create histograms.

To measure the content of DNA in FH cells and determine their ploidy 1 × 10^6^ of dissociated cells were fixed overnight in 70% ethanol at −20 °C. The next day, cells were centrifuged at 300× *g* at RT, resuspended in 1 mL PBS and treated with RNAse A (Invitrogen) at the final concentration of 100 μg/mL for 15–20 min at 37 °C. The suspension was then equally distributed into two 1.5 mL microcentrifuge tubes. One tube received propidium iodide (PI) at a final concentration of 50 μg/mL, and the other was maintained as a control without PI. The samples were incubated on ice for 15 min. The DNA content of cells was analyzed using FACScan in the FL3-A channel. Dead cells and aggregates were gated out based on forward scatter, FL3-A, and FL3-W parameters. Tetraploid FH cells, diploid parental HM-1 ESCs, and splenocytes were included in the analysis.

### 4.7. Immunocytochemistry

For detection of cardiac structural proteins, beating areas in EBs of HM-1 ESCs and FH cells were microdissected with a scalpel at day 14 and replated on 0.1% gelatine-coated coverslips. They were fixed at day 21 with 100% ice-cold methanol and rehydrated with PBS. For analysis of MHC class I expression, microdissected beating areas were dissociated with 0.25% trypsin/0.53 mM EDTA for 10 min at 37 °C and single cells were plated on a 0.1% gelatine-coated µ-Slide 8-well chambered coverslip (ibidi). After two more days in culture, cells were treated with 100 ng/mL interferon γ (IFNγ) for 48 h to increase the levels of MHC class I molecules on the cell surface. IFNγ-treated cells were fixed with 4% paraformaldehyde. Immortalized C57/SV and Balbc/SV fibroblasts were used as a positive control for stainings with H-2K^b^ and H-2K^d^ antibodies, respectively. Blocking was carried out with 5% bovine serum albumin (BSA, VWR International, Langenfeld, Germany) in PBS for 1 h. Incubations with anti-sarcomeric-α-actinin (Sigma-Aldrich, clone EA53, 1:400), anti-cardiac troponin T (Labvision, Fremont, CA, USA; rabbit polyclonal, 1:100), anti-H-2K^b^ (BD Pharmingen, clone AF6-88.5, 1:50) or H-2K^d^ (BD Pharmingen, clone SF1-1.1, 1:50) were undertaken overnight at 4 °C in 1% BSA in PBS. Samples were incubated with secondary antibodies anti-mouse-IgG2a-AlexaFluor647 (BD Pharmingen, 1:1000) or anti-mouse-IgG1-AlexaFluor488 (Invitrogen, 1:1000) for 90 min at room temperature. Nuclei were stained using Hoechst 33342 (2 µg/mL, Sigma-Aldrich). After washing, samples were kept until analysis in DABCO or embedded in ProlongGold Antifade Reagent (Invitrogen). Images were acquired on a Zeiss Axiovert 200M fluorescence microscope (Zeiss) and analyzed with Zeiss Axiovision 4.5 software.

### 4.8. Teratoma Assay

For induction of teratomas, FH cell colonies were enzymatically dissociated with 0.05% trypsin/0.53 mM EDTA (Invitrogen) and reconstituted in PBS to a final concentration of 2 × 10^6^ cells/mL. Until injection, cells were stored on ice. The cell suspension was injected into the hind limbs of narcotized (Isoflurane) immunodeficient SCID/beige mice. 1 × 10^6^ cells were injected into the left and 2 × 10^5^ cells into the right hind limb. Mice were regularly monitored for tumor occurrence. After 4 weeks, mice were sacrificed, and the tumors were excised and fixed with 4% PFA overnight. Fixed tumors were embedded in paraffin, sliced, and analyzed by H&E staining, Alcian blue staining to identify goblet cells and cartilage, and by staining with antibodies specific for cell lineages derived from each of the three primary germ layers: endoderm (cytokeratin 8 in secretory epithelial cells), mesoderm (α-actinin in CMs) and ectoderm (βIII-tubulin in neurons).

### 4.9. SNP Genotyping

We used sequencing of single nucleotide polymorphisms (SNP) to confirm the presence of variant alleles in clones of FH cells, each originating from a genome of a different mouse strain—129/Ola (source of ESCs) and DBA/2J (source of somatic cells), as described by us previously [97]. The SNPs were identified by a genome-wide screen of the SNP variation between these two strains of mice to identify single nucleotide variations on each of the chromosomes. The database for Mouse Genome Informatics was used: http://www.informatics.jax.org/javawi2/servlet/WIFetch?page=snpQF (accessed on 18 July 2014). SNP sites located in the coding regions of each chromosome were randomly selected, and primers were designed in order that 200 bases spanning the 3′ and 5′ ends of the variant nucleotide could be amplified. In this way, 21 primer pairs were designed to recognize a single sequence with one allelic variation for each of the 21 chromosomes of the mouse genome (19 autosomes and each of two gonosomes). Genomic DNA was isolated from HM-1 ESCs, splenocytes, and FH cells, using the DNeasy Blood and Tissue kit (Qiagen, Hilden, Germany), and PCR was performed with 100 ng of DNA to amplify the sequence encompassing the specific SNP using primers listed in Appendix A at a final concentration of 0.15 μM. PCR products pooled from 4–5 PCR reactions were purified using the QIAquick PCR Purification kit (Qiagen) and then sequenced in the central facility of the Centre for Molecular Medicine Cologne (CMMC). Sequences were analyzed for the presence of heterozygous nucleotides at the expected SNP in FH cells using Chromas Pro1.5 software.

### 4.10. RT-PCR Analysis

For reverse transcriptase-polymerase chain reactions (RT-PCR), PSCs or embryoid bodies (EBs) were lysed in Trizol Reagent (Invitrogen) to isolate total RNA. RNA was DNAse treated (DNAse I, amplification grade, Invitrogen), and 1 µg total RNA was transcribed into cDNA using random hexamers and Superscript II Reverse transcriptase (Invitrogen) according to the manufacturer’s protocol. PCR was performed using the Jump Start Red Taq polymerase kit (Sigma-Aldrich). Primers used for the amplification of target sequences are listed in Appendix A. After initial denaturation at 94 °C for 2 min, all amplifications were carried out using denaturation at 94 °C for 35 s, annealing at 56–60 °C for 45 s, and elongation at 72 °C for 75 s. For each reaction, 2% of the cDNA initially generated was used as a template. After 32–35 cycles, the final elongation was carried out at 72 °C for 5 min. PCR fragments were separated on 2% agarose gels (UltraPure Agarose, Invitrogen) containing ethidium bromide.

### 4.11. Patch-Clamp

We used the standard whole-cell patch-clamp recording technique for the electrophysiological characterization of ES and FH cell-derived CMs [98]. To compare the maturation process among the lines, differentiating CM were functionally characterized on days 16–19 after initiation of differentiation. For patch-clamp experiments, individual CMs were selected according to their typical morphology and spontaneous beating activity. The APs were classified based on their typical shape and duration, as characterized in earlier work of our group for mouse pluripotent cell-derived CMs [74,81]. The ventricular-like subtype is also characterized by the presence of a plateau phase, and the atrial-like subtype by its triangular shape. The quantitative aspects are reflected in the APD90 and the APD90/APD50 values, which yield clear differences between ventricular-like CMs to the other two CM subtypes. All recordings were performed using an EPC9 amplifier and operated through the PULSE acquisition software (HEKA, Reutlingen, Germany). The glass coverslips containing the cells were placed into a temperature-controlled (37 °C) recording chamber and perfused continuously with extracellular solution. Cell membrane capacitance was determined online using the PULSE program.

In the current-clamp experiments, we recorded typical action potentials (AP) of different types of CMs. The extracellular solution contained (in mM): NaCl 140, KCl 5.4, CaCl_2_ 1.8, MgCl_2_ 1, glucose 10, 4-(2-hydroxyethyl)-1-piperazine-ethanesulfonic acid (HEPES) 10, pH adjusted to 7.40 at 37 °C with NaOH. The intracellular solution contained (in mM): KCl 50, K-Aspartate 80, MgCl_2_ 1, MgATP 3, glycol-bis (2-aminoethylether)-N, N, N, N-tetraacetic acid (EGTA) 10, and HEPES 10, pH adjusted to 7.40 with KOH. We additionally tested the CM for functional muscarinic and β-adrenergic regulation by applying 1 µM carbachol (Cch) or 1 µM isoproterenol (Iso), respectively. We also examined the effects of Na^+^ (30 µM lidocaine), L-type Ca^2+^ (1 µM nifedipine), and mERG (250 nM E4031) channel blockers.

In the voltage-clamp experiments, voltage-gated Na^+^ and L-type Ca^2+^ channel currents were recorded. The extracellular solution contained (in mM): NaCl 120, KCl 5, CaCl_2_ 3.6, MgCl_2_ 1, tetraethyl ammonium (TEA) chloride 20, and HEPES 10, pH adjusted to 7.40 at 37 °C with TEA-OH. The intracellular solution contained (in mM): CsCl 120, MgCl_2_ 3, MgATP 5, EGTA 10, and HEPES 5, pH adjusted to 7.40 with CsOH. All substances were from Sigma-Aldrich.

The data on L-type Ca^2+^ currents and Na^+^ currents were collected simultaneously. There was no contamination of L-type Ca^2+^ currents with Na^+^ currents, as the latter was fully inactivated with a long conditioning pre-pulse to –40 mV before the L-type Ca^2+^ currents were elicited by a test pulse. In addition, L-type currents were always checked for the presence of fast activation component, which would suggest contamination with Na^+^ current; there was no such component visible. L-type Ca^2+^ channels were not affected by the pre-pulse, as they activate in a more positive voltage range. We also estimated the Ca^2+^ current contamination in Na^+^ currents. As L-type currents were always co-measured in the cells where Na^+^ channel currents were recorded, we compared the amplitudes. If the maximal amplitudes at corresponding potentials are compared, the mean contamination of L-type currents is less than 1%; in the most important voltage range (from –50 to –10 mV), it is less than 0.5%. However, the activation of L-type Ca^2+^ channels is 5–10 times slower than the Na^+^ channel activation, reducing the L-type current contribution to <0.1%. Therefore, the Ca^2+^ current contamination in Na^+^ currents could be neglected.

In all whole-cell recordings, leak subtraction was applied. PULSE software, Excel (Microsoft, Redmond, WA, USA), Sigma Plot (Systat Software GmbH, Erkrath, Germany), CorelDraw (Corel GmbH, Munich, Germany), and Origin (OriginLab Corporation, Northampton, MA, USA) programs were used to display and analyze recordings.

### 4.12. Statistical Analysis

Statistical analysis of drug effects was performed by one-way ANOVA followed by the post hoc Bonferroni’s test to correct for multiple comparisons. Comparison of AP parameters between cell lines (HM-1 vs. FH2.1 vs. FH4.2) and between CM-subtypes (P-like vs. A-like vs. V-like) was performed by two-way ANOVA followed by the post hoc Tukey’s test to correct for multiple comparisons. A *p*-value lower than 0.05 was considered to be statistically significant.

## Figures and Tables

**Figure 1 ijms-24-06546-f001:**
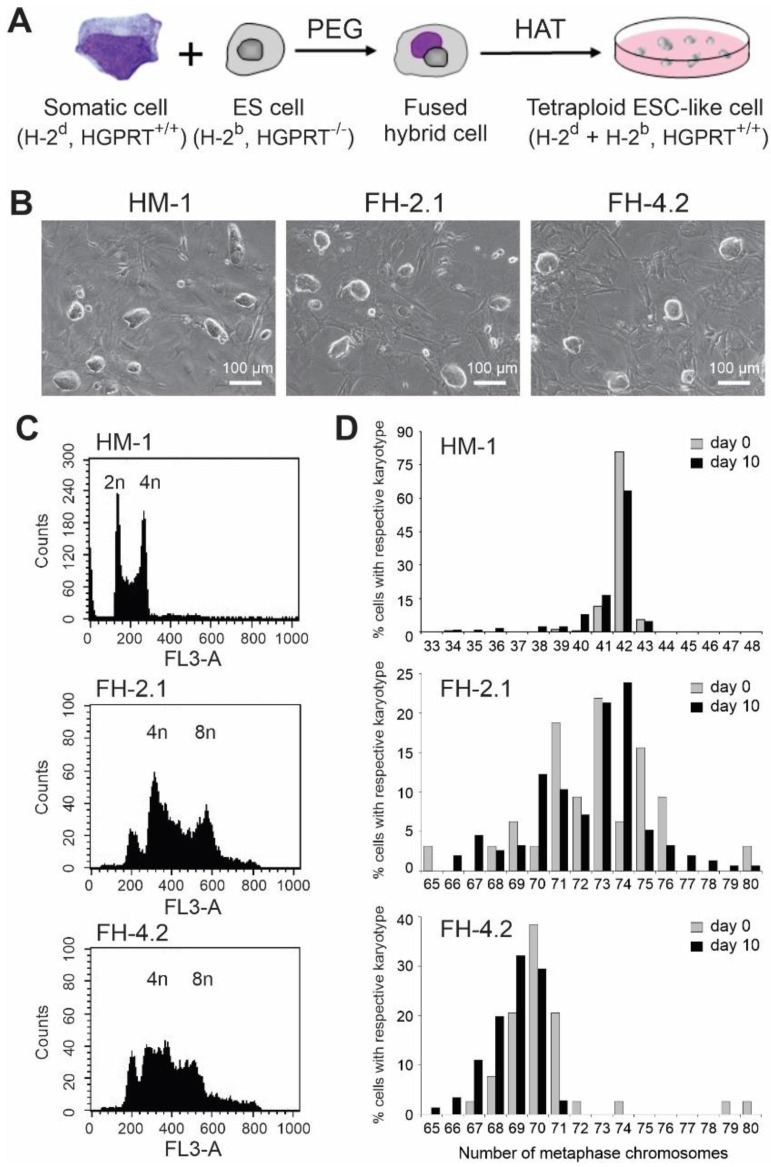
Characterization of fusion-derived hybrid (FH) cells. (**A**) Scheme summarizing the process of fusion-based reprogramming of bone marrow cells and splenocytes using PEG and subsequent selection with HAT to remove unfused embryonic stem cells (ESCs). Fusion partners were selected from two different mouse strains to be distinguishable by their MHC class I haplotypes and SNPs. (**B**) Colonies of the original HM-1 ESC line, bone marrow cell-derived (FH-2.1), and splenocyte-derived (FH-4.2) fusion clones. Scale bars: 100 μM. (**C**) DNA content analysis of undifferentiated ESCs and fusion-hybrid-derived pluripotent stem cells (FH-PSCs) as determined by flow cytometry of propidium iodide stained cells. (**D**) Karyotype analysis of undifferentiated ESCs and FH-PSCs and their respective differentiating cells isolated from embryoid bodies (EBs) at day 10 of differentiation (see also Appendix A for a representative spread of metaphase chromosomes).

**Figure 2 ijms-24-06546-f002:**
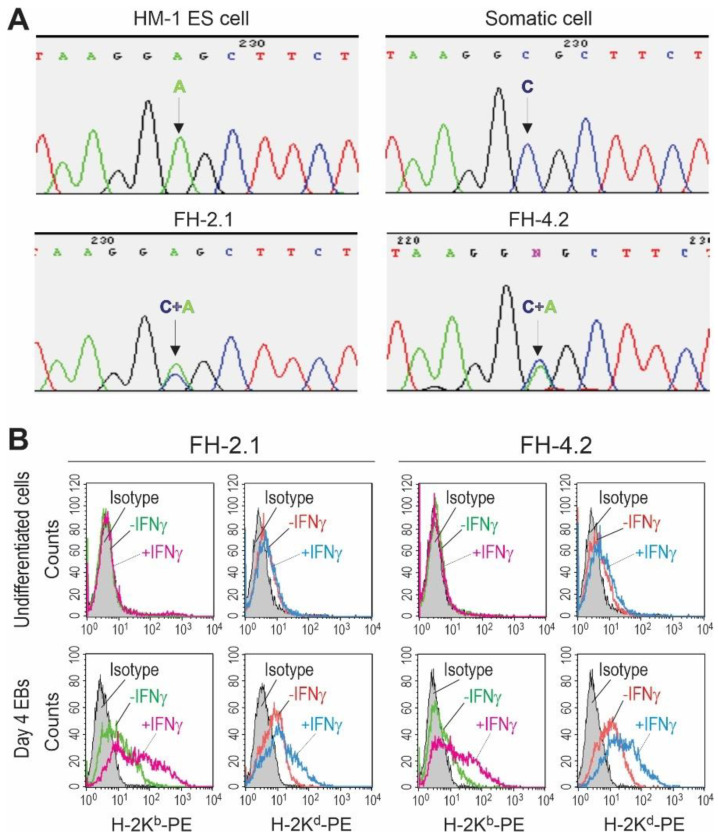
Identification of both parental genomes in fusion-hybrid-derived PSCs. (**A**) Representative example of identification of both parental genomes in FH-PSCs using single nucleotide polymorphism (SNP) genotyping. The DNA locus encompassing the selected SNP on chromosome 7 was amplified by PCR and sequenced. Only one polymorphic allele could be detected in HM-1 ESCs (adenine, A) and somatic cells (cytosine, C). In contrast, FH-PSC lines contained both variant alleles (C/A) at this SNP position, suggesting the existence of both parental genomes in these cells. Complimentary analyses were performed for all autosomes and are summarized in Appendix A. (**B**) Expression of major histocompatibility (MHC) class I molecules on the surface of fusion-derived PSCs FH-2.1 and FH-4.2. The levels of MHC class I molecules of parental ESCs (H-2K^b^) and somatic cells (H-2K^d^) were determined on intact (−IFNγ) or interferon-gamma-treated (+IFNγ) (100 ng/mL, 2 days) FH-PSCs at undifferentiated (upper panels) and differentiating state (day 4 EBs, lower panels) by flow cytometry.

**Figure 3 ijms-24-06546-f003:**
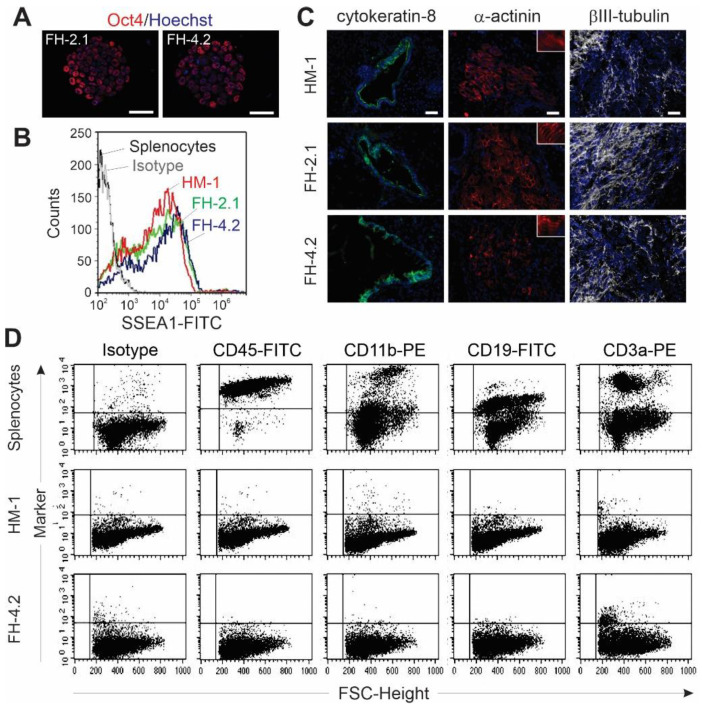
Successful reprogramming of FH cells to a pluripotent state. (**A**) ESC-like colonies of fusion-derived hybrid cells FH-2.1 and FH-4.2 express the pluripotency marker Oct-4. (**B**) Flow cytometric analysis of SSEA-1 expression on HM-1 ESCs, FH-PSC lines FH-2.1, and FH-4.2, as well as on splenocytes, were used as a negative control. (**C**) FH-derived PSCs form teratoma in vivo after subcutaneous injection into SCID/beige mouse hind legs. Cells of all three germ layers could be demonstrated by staining with antibodies specific for endoderm (cytokeratin-8, green), ectoderm (βIII-tubulin, white), and mesoderm (α-actinin, red) (see also Appendix A for histological analysis of teratoma derived from the FH-4.2 cell line). The nuclei were counter-stained with DAPI (blue). Scale bars: 50 μm. (**D**) Expression of somatic cell markers of lymphoid lineage (CD45, CD11b, CD19, and CD3a) on splenocytes, HM-1 ESC, and splenocyte-derived FH-4.2 PSCs, as determined by flow cytometry.

**Figure 4 ijms-24-06546-f004:**
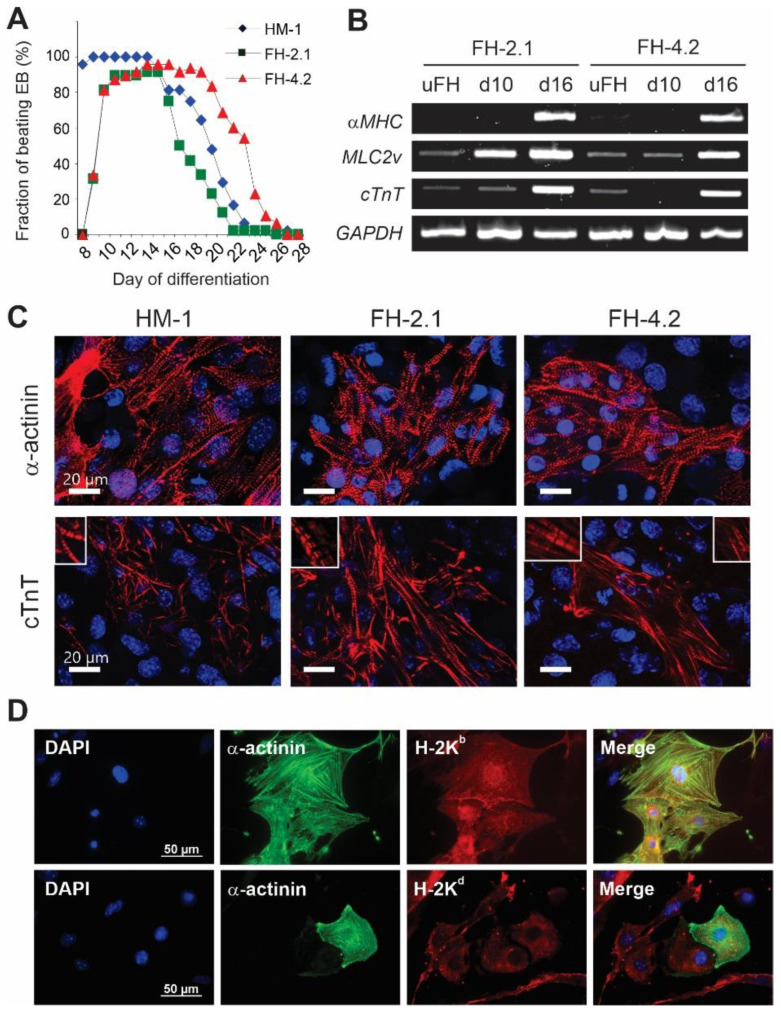
Cardiac differentiation of fusion hybrid PSCs. (**A**) The percentage of spontaneously beating EBs was calculated at the indicated time points during differentiation. (**B**) Semiquantitative RT-PCR analysis of expression of cardiac-specific genes *αMHC*, *MLC2v*, and *cTnT* in EBs derived from FH-2.1 and FH-4.2 cells collected at day 10 (d10) and day 16 (d16) of differentiation. Undifferentiated FH cells (uFH) were included as controls, and GAPDH was used as a housekeeping gene. (**C**) Immunocytochemical detection of α-actinin (upper panels) and cardiac troponin T (cTnT, lower panels) in beating areas derived from HM-1 ESCs, and FH-2.1 and FH-4.2 PSCs. Cardiac markers are shown in red. Scale bars: 20 μM. (**D**) Immunocytochemical detection of α-actinin (green) and MHC class I molecules H-2K^b^ and H-2K^d^ (red) on FH-4.2 cells at day 16 of differentiation. Prior to staining, beating areas were trypsinized, single cells were plated on fibronectin-coated cover slides and treated for 2 days with IFNγ (100 ng/mL) to facilitate the detection of MHC class I molecules. These results demonstrate that all FH-PSC-derived cells, including CMs (green), express MHC class I molecules (red) of both fusion partners. Nuclei in all images were counter-stained with DAPI (blue). Scale bars = 50 μM. The results of the validation of the specificity of antibodies used for the detection of H-2K^b^ and H-2K^d^ molecules are shown in Appendix A.

**Figure 5 ijms-24-06546-f005:**
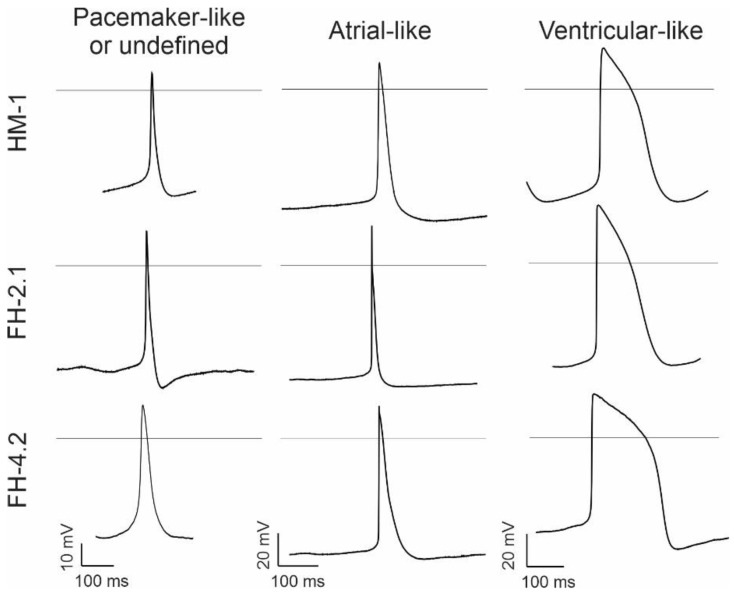
Action potential (AP) analysis. Typical APs of different CM subtypes were recorded from CMs derived from HM-1 ESCs and FH cells (clones FH-2.1 and FH-4.2) at days 16–19 of differentiation. Detailed analysis of different AP parameters in these cells is given in Table 1 and Appendix A.

**Figure 6 ijms-24-06546-f006:**
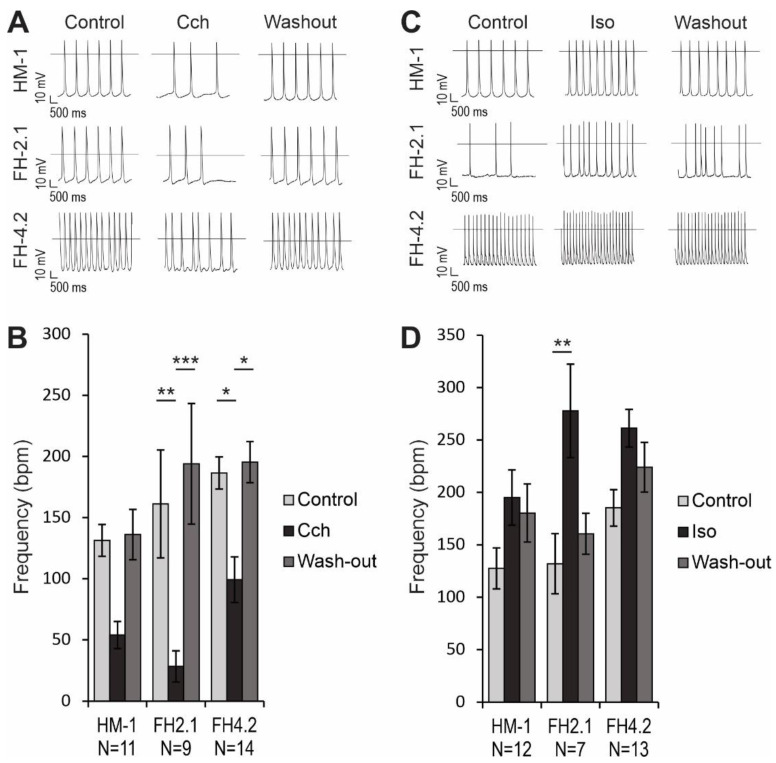
Effect of carbachol (CCh) and isoproterenol (Iso) on HM-1-ESC- and FH-PSC-derived CMs. Negative chronotropic effect of 1 μM CCh (**A**,**B**) and 1 μM Iso (**C**,**D**) on HM-1 CMs, FH-2.1 CMs, and FH-4.2 CMs isolated at days 16–19 of differentiation. APs were recorded before the application of drugs (Control), in the presence of a drug (CCh or Iso), and after washout of the applied drug (Washout). Representative traces are shown in panels A (for Cch) and C (for Iso). Quantitative analysis of the effect of drugs on AP frequency in HM-1 ESC- and FH-PSC-derived CMs is shown in panels B (for Cch) and D (for Iso). Data are presented as the mean ± SEM of N cells analyzed. Statistical analysis was performed by one-way ANOVA followed by the post hoc Bonferroni’s test to correct for multiple comparisons. *p*-values are only given for group comparisons with statistically significant differences. * *p* < 0.05, ** *p* < 0.01, *** *p* < 0.001.

**Figure 7 ijms-24-06546-f007:**
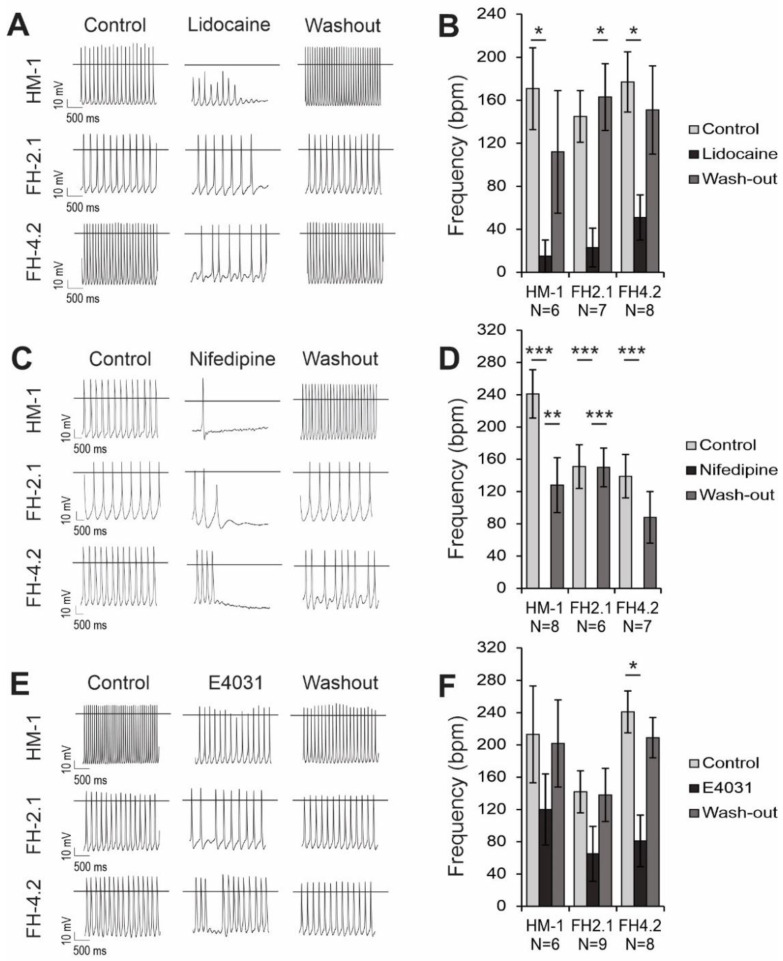
Effect of lidocaine, nifedipine, and E4031 on ESC- and FH-PSC-derived CMs. (**A**,**C**,**E**) APs were recorded before the application of drugs (Control), in the presence of 30 μM lidocaine, 1 μM nifedipine, or 0.25 μM E4031, and after washout of drugs (washout). Representative traces are shown. (**B**,**D**,**F**) Statistical evaluation of the drug effect on beating frequency in HM-1 ESC-CMs, FH-2.1 CMs, and FH-4.2 CMs. Data are presented as the mean ± SEM. N indicates the number of cells analyzed. Statistical analysis was performed by one-way ANOVA followed by the post hoc Bonferroni’s test. *p*-values are only given for group comparisons with statistically significant differences. * *p* < 0.05, ** *p* < 0.01, *** *p* < 0.001.

**Figure 8 ijms-24-06546-f008:**
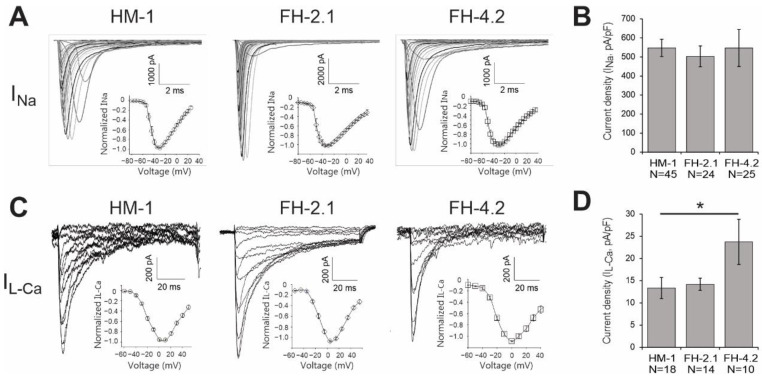
Functional expression of voltage-gated Na^+^ and Ca^2+^ ion channels in ESC- and FH-PSC-derived CMs. (**A**) Na^+^ channel currents were elicited by a family of 100 ms depolarizations from a −90 mV holding potential (HP) to voltages ranging from −60 to −55 mV in 5 mV steps. (**C**) Ca^2+^ channel currents were measured with a double-pulse protocol. First, a 100 ms depolarization from an HP of −90 mV to −40 mV was applied to inactivate Na^+^ and T-type Ca^2+^ channels, and then L-type Ca^2+^ channels were elicited by a family of 67 ms depolarizations to voltages ranging from −60 to −50 mV in 10 mV steps. In Na^+^ channel (**A**) and Ca^2+^ channel (**C**) experiments, leak subtraction P/4 protocol was applied from HP of −90 mV. Insets in (**A**,**C**) show the current–voltage (I–V) relationships of I_Na_ and I_CaL_, respectively. Peak Na^+^ (**B**) and L-type Ca^2+^ current densities (**D**) in HM-1 ESC-CMs, FH-2.1 CMs, and FH-4.2 CMs were determined at −40 mV and 0 mV, respectively. Data are presented as the mean ± SEM. N indicates the number of cells analyzed. Statistical analysis was performed by one-way ANOVA followed by the post hoc Bonferroni’s test. *p*-values are only given for group comparisons for which the differences were statistically significant. * *p* < 0.05.

**Table 1 ijms-24-06546-t001:** Action potential (AP) parameters for different subtypes of CMs derived from diploid murine HM-1 ESCs and near-tetraploid fusion-derived hybrid PSCs (clone FH-2.1 and FH-4.2) ^1^.

CM Origin	AP Type	Peak (mV)	MDP (mV)	Height (mV)	Vdd (mV/s)	Frequency (1/min)	Vmax (V/s)	APD90 (ms)	APD90/ APD50	Cells (N)	Cells (%)
HM-1	UD	23 ± 9	−49 ± 5	72 ± 8	0.06 ± 0.01	136 ± 27	13 ± 4	83 ± 14	2.7 ± 0.8	7	26
	A	29 ± 4	−53 ± 5	82 ± 7	0.06 ± 0.02	168 ± 39	21 ± 4	86 ± 15	2.5 ± 0.3	6	22
	V	36 ± 2	−56 ± 2	90 ± 3	0.07 ± 0.01	163 ± 24	23 ± 2	136 ± 13	1.6 ± 0.1	14	52
FH-2.1	UD	22 ± 3	−41 ± 4	63 ± 6	0.06 ± 0.01	125 ± 14	6 ± 1	100 ± 23	2.4 ± 0.2	8	33
	A	38 ± 6	−52 ± 2	90 ± 5	0.03 ± 0.01	167 ± 36	21 ± 2	86 ± 14	3.7 ± 0.8	11	46
	V	36 ± 1	−54 ± 4	93 ± 3	0.03 ± 0.02	118 ± 45	21 ± 4	122 ± 14	1.5 ± 0.1	5	21
FH-4.2	UD	25 ± 2	−49 ± 2	74 ± 2	0.11 ± 0.02	190 ± 42	8 ± 2	79 ± 10	3.8 ± 1.3	8	33
	A	25 ± 4	−57 ± 2	83 ± 4	0.07 ± 0.01	182 ± 16	21 ± 2	77 ± 8	3.4 ± 0.6	13	54
	V	29 ± 4	−48 ± 14	78 ± 18	0.07 ± 0.01	119 ± 18	12 ± 3	194 ± 74	1.5 ± 0.1	3	13

^1^ We estimated the maximal diastolic potential (MDP), AP frequency, maximum velocity of depolarization (Vmax), velocity of diastolic depolarization (Vdd), AP duration at 90% repolarization (APD90), and ratio of APD90/APD50, where APD50 represents AP duration at 50% repolarization. Cardiomyocytes were categorized according to their AP types as undefined (UD, some of which might represent pacemaker-like cells), ventricular-like (V), and atrial-like (A). Data are shown as mean ± standard error of the mean (SEM) of N cells analyzed. A detailed statistical analysis of these results is presented in Appendix A.

## Data Availability

Data is available upon reasonable request to the corresponding author.

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
