# Peer review of "Electrophysiological Properties of Tetraploid Cardiomyocytes Derived from Murine Pluripotent Stem Cells Generated by Fusion of Adult Somatic Cells with Embryonic Stem Cells"

_ijms, 2023, doi:10.3390/ijms24076546_

Round 1

Reviewer 1 Report

In this paper, the authors describe reprogramming of murine somatic cells by fusion with mESCs, to generate near-tetraploid cells, and the comparison of these cells to control diploid mESC cells. They show that besides a minor difference in maturation rate, the cells exhibit similar characteristics as pluripotent colonies and as differentiated cardiomyocytes driven from EBs. The authors prove, using multiple methods,  that the cells are tetraploid that contain DNA from both original cells. In addition, their function as cardiomyocytes is thoroughly assessed. The paper is clearly written and the methods support the conclusions. I guess that authors would have preferred to more significant differences between diploid and mutiploid cardiomyocytes, as they found only slight differences (tendency of atrial action potential in the multiploid cells), however I think this paper is of interest to the relevant community. 

  minor comments:  please add arrows in Figure S5 to highlight the tissue you are describing. 

Author Response

We thank this reviewer for taking the time to assess the merit of our work and to give her/his favorable opinion about our study and the relevance of our findings. We have responded to the concern raised by the reviewers to the best of our ability. Figure S5 was updated and asterisks were added to highlight the histological structures mentioned in the figure legend. All changes made in the revised version of the manuscript are highlighted in red. We hope that the revised manuscript will be found acceptable for publication in its present form. 

Reviewer 2 Report

Major Comment:  this is an interesting paper that attempts to study polyploid CM and the electrophysiological consequences of polyploidy in mouse in fused cardiomyocytes. If the goal was to study the electrophysiological differences between diploid and polyploid cells, then it is puzzling why the authors did not choose to test for the differences in isolated adult mice myocyte, where these cell types are generally expressed in the mouse heart. Instead, the authors used a fusion approach which might be fraught with technical and cytological issues. At the minimum studding the ploidy in native adult myocytes could have served as the control for their fusion experiments. I think it would be useful to obtain such information which is not available to my awareness; not counting studies on “mononuclear” that could be diploid or tetraploid. Alternatively, they could have taken diploid ES or iPSCs, induce differentiation, treat at the appropriate time with any of several things to block mitosis, and thus obtain real polyploid CMs. This study is mostly technology in search of a purpose. None of the rationalizations they give in the discussion or elsewhere have great validity.

Minor Concerns:

1)Electrophysiologically, I would have liked to see quantification of membrane capacitance, to know how big are the fused cells?

2)If the fusing membranes lack Ca2+ or Na+ channels, why then are the resultant cells not showing lower density of channels?

3)Why did the authors not measure HCN encoded If channel currents, especially since the cells seem to beat spontaneously and respond to adrenergic and cholinergic agonists?

4)In electrophysiological classification of cells into atrial/ ventricular/pacemaking cells, did the authors have any biochemical/genetic data that corroborated their electrophysiological classification? Such studies would greatly improve the manuscript.

Author Response

We thank the reviewer for taking the time to assess the merit of our work and for her/his valuable comments. We have responded to the concerns raised by the reviewer to the best of our ability. Our specific responses to the reviewers’ criticism are listed below. All changes made in the revised version of the manuscript are highlighted in red. We are grateful to the reviewer’s insights into our findings and hope that the revised manuscript will be found acceptable for publication in its present form. 

Reply to the major comment:

We thank the reviewer for her/his comments and constructive suggestions. We agree that studying the effect of ploidy on electrophysiological properties of native adult cardiomyocytes (CMs) would have served as a good control for our fusion experiments. Having said this, the identification of the nuclearity and in particular ploidy status of individual CMs immediately after isolating the cells is challenging and currently to the best of our knowledge not yet fully established. We have therefore chosen an alternative approach by using the reported fusion approach. Thus, although there is no doubt that analyses with freshly isolated CMs from hearts of mice as well as humans would be very interesting and should be performed in future studies, we believe that the data reported in our manuscript still have merit on their own. We also concur that our findings cannot be directly extrapolated to native CMs, also because the electrophysiological properties of native adult CMs and CMs derived from ESCs or iPSCs differ significantly due to different levels of their structural and functional maturity (Schmid C, et al., Cells 2021, DOI: 10.3390/cells10123370). However, this does not invalidate the main conclusion of our study that electrophysiological properties of in vitro derived mESC-CMs are not affected by their ploidy, which is fully supported by the data presented.

By choosing the cell fusion technology and the ESC model to generate polyploid CMs, we established an in vitro controlled system that enabled us to generate homogeneous populations of these cells at large quantities not only for studying their electrophysiology, but also for future studies of their properties at different stages of differentiation and under different experimental conditions. This would not be possible with CMs isolated from adult hearts. Establishing the validity of this system in a murine model also lays ground for similar studies in a human system, because primary CMs harvested from healthy or diseased hearts are not easily accessible.

As this reviewer has pointed out, an alternative approach to obtaining large numbers of true polyploid CMs from ESCs or iPSCs would be to treat the differentiating cells with drugs that block mitosis such as microtubule dissociating reagents colchicine and nocodazole (Grove LE and Ghosh RN, Assay Drug Dev Technol 2006, DOI: 10.1089/adt.2006.4.421). Although cells artificially driven into tetraploidy by blocking mitosis may not be fully comparable to native tetraploid CMs, this is certainly a valid suggestion and we will consider performing these analyses in our follow-up experiments to complement the data obtained from fused cells.

Reply to the minor comments:

1) Electrophysiologically, I would have liked to see quantification of membrane capacitance, to know how big are the fused cells.

We thank the reviewer for bringing up this important question. We have determined the membrane capacitance and the results of these measurements are now provided in the new Figure S9 and mentioned in the Results section. Although there is a tendency towards higher capacitance in near-tetraploid cells compared to diploid ones, this data show that membrane capacitance of CMs derived from diploid HM1 mESCs and near-tetraploid fusion-derived FH2.1 and FH4.2 mESCs do not differ significantly.

In general, increased ploidy is associated with an increase in cell size as well as mRNA and protein abundance (Pandit SK, et al., Trends Cell Biol 2013, DOI: 10.1016/j.tcb.2013.06.002). However, in our experimental model, we do not observe clear effects. One possibility is that a clear effect of ploidy on cell size may have been masked by the presence of different CM subtypes as well as CMs at different stages of maturation in these cultures (Schmid C, et al., Cells 2021, DOI: 10.3390/cells10123370) potentially contributing to the observed variability in membrane capacitance.

2) If the fusing membranes lack Ca2+ or Na+ channels, why then are the resultant cells not showing lower density of channels?

We thank the reviewer for this comment. The fusing membranes of parental mESCs and somatic cells (splenocytes or bone marrow cells) most likely do not express voltage-activated cardiac Ca2+ or Na+ channels. The fusion-derived polyploid pluripotent stem cell clones were established as stable cell lines by subsequent culturing of fused cells in the HAT-supplemented medium to eliminate unfused mESCs. These hybrid ESC clones were subjected to in vitro differentiation into CMs. Within the time frame of 16-19 days, mESC underwent significant epigenetic, transcriptional, structural, and also functional changes that led to the emergence of cardiac myocytes. In these cells, global gene expression was adjusted to the needs of the new cell type and, most likely, to their DNA content and cell size (Marguerat S and Bähler J, Trends Genet 2012, DOI: 10.1016/j.tig.2012.07.003). Therefore, we assume that the membranes of the parental fusion partners underwent radical changes during the reprogramming of the somatic genome into the pluripotent state and during the subsequent differentiation of ESCs toward CMs.

3) Why did the authors not measure HCN encoded If channel currents, especially since the cells seem to beat spontaneously and respond to adrenergic and cholinergic agonists?

Cultures of ESC- and iPSC-derived CMs are mostly comprised of spontaneously beating CMs although some quiescent CMs might also be present. However, due to their developmental immaturity, the spontaneous beating does not serve as a reliable marker for distinction between different CM subtypes (Schmid C, et al., Cells 2021, DOI: 10.3390/cells10123370). In our study, we measured only cells that showed spontaneous beating activity under the microscope. Since initially Ca++ and later Na+ currents were responsible for the fast upstroke (with the exception of pacemaker-like cells), we have focused on these key conductances. This focus is also explained by the fact that If has been reported to be expressed in early cardiomyocytes regardless of the subtype but subsequently downregulated (Yechikov et al., Stem Cells 2016, DOI: 10.1002/stem.2466).

4) In electrophysiological classification of cells into atrial/ ventricular/pacemaking cells, did the authors have any biochemical/genetic data that corroborated their electrophysiological classification? Such studies would greatly improve the manuscript.

We thank the reviewer for pointing out this important issue. It is possible that additional analyses, such as a subsequent single-cell gene expression analysis, could have provided some additional confirmation for our electrophysiological classification of the measured cells. However, it is unclear whether additional biochemical or genetic data would help to identify different CM subtypes unequivocally. For example, Yechikov and coworkers have correlated the action potential profiles of individual pacemaker-, atrial- and ventricular-like hiPSC-CMs with the expression of the proposed pacemaker-specific markers HCN4 and Isl1 at the protein level in the same cell (Yechikov et al., Stem Cells 2016, DOI: 10.1002/stem.2466). They found that these two markers were expressed in all three hiPSC-CM subtypes. Therefore, these markers alone are not sufficient to identify hiPSC-derived pacemaker-like CMs. In another study, the group of Timothy Nelson has used single-cell RNA-seq and shown that individual cellular expression patterns alone are not able to categorize the individual hiPSC-CMs into atrial, ventricular, or nodal subtypes as determined by electrophysiology measurements (Biendarra-Tiegs et al., Stem Cells Dev 2019, DOI: 10.1089/scd.2019.0030). Most recently, Schmid and coworkers used electrophysiological and single-cell RT-qPCR data from the same cell to determine if ion channel transcripts correlate with the electrophysiological parameters. They showed that the majority of individual cells in three different commercially available hiPSC-CM preparations do not represent chamber-specific cell populations present in the adult human heart and exhibit unexpected combinations of ion channel transcripts and electrophysiological parameters on the single-cell level, combining characteristics of nodal, atrial and ventricular CMs (Schmid C, et al., Cells 2021, DOI: 10.3390/cells10123370).  Although we believe that such analyses would certainly add valuable information to our existing data, they may not be conclusive enough to confirm our electrophysiological classification of mESC-CMs and go in our view beyond the scope of the present manuscript. We have incorporated some of these important aspects in the Discussion section of the manuscript.

References cited in this reply:

  1. Schmid C, Abi-Gerges N, Leitner MG, Zellner D, Rast G. Ion Channel Expression and Electrophysiology of Singular Human (Primary and Induced Pluripotent Stem Cell-Derived) Cardiomyocytes. Cells. 2021; 10(12):3370. https://doi.org/10.3390/cells10123370. PMID: 34943878.
  2. Grove LE, Ghosh RN. Quantitative characterization of mitosis-blocked tetraploid cells using high content analysis. Assay Drug Dev Technol. 2006 Aug;4(4):421-42. doi: 10.1089/adt.2006.4.421. PMID: 16945015.
  3. Pandit SK, Westendorp B, de Bruin A. Physiological significance of polyploidization in mammalian cells. Trends Cell Biol. 2013;23(11):556-66. doi: 10.1016/j.tcb.2013.06.002. PMID: 23849927
  4. Marguerat S, Bähler J. Coordinating genome expression with cell size. Trends Genet. 2012 Nov;28(11):560-5. doi: 10.1016/j.tig.2012.07.003. PMID: 22863032.
  5. Yechikov S, Copaciu R, Gluck JM, Deng W, Chiamvimonvat N, Chan JW, Lieu DK. Same-Single-Cell Analysis of Pacemaker-Specific Markers in Human Induced Pluripotent Stem Cell-Derived Cardiomyocyte Subtypes Classified by Electrophysiology. Stem Cells. 2016;34(11):2670-2680. doi: 10.1002/stem.2466. PMID: 27434649.
  6. Biendarra-Tiegs SM, Li X, Ye D, Brandt EB, Ackerman MJ, Nelson TJ. Single-Cell RNA-Sequencing and Optical Electrophysiology of Human Induced Pluripotent Stem Cell-Derived Cardiomyocytes Reveal Discordance Between Cardiac Subtype-Associated Gene Expression Patterns and Electrophysiological Phenotypes. Stem Cells Dev. 2019;28(10):659-673. doi: 10.1089/scd.2019.0030. PMID: 30892143.